# When does selection favor learning from the old? Social learning in age-structured populations

Dominik Deffner[1,2,3]*, Richard McElreath[1]

**1** Department of Human Behavior, Ecology and Culture, Max Planck Institute for Evolutionary Anthropology, Leipzig, Germany, **2** Science of Intelligence Excellence Cluster, Technical University Berlin, Berlin, Germany, **3** Center for Adaptive Rationality, Max Planck Institute for Human Development, Berlin, Germany

* deffner@mpib-berlin.mpg.de

**Data Availability Statement:** This article does not contain any new data. R simulation and plotting code, as well as Wolfram Mathematica analysis notebook can be found on GitHub: https://github.

## Abstract

Culture and demography jointly facilitate flexible human adaptation, yet it still remains unclear how social learning operates in populations with age structure. Here, we present a mathematical model of the evolution of social learning in a population with different age classes. We investigate how demographic processes affect the adaptive value of culture, cultural adaptation and population growth, and identify the conditions that favor learning from older vs. younger individuals. We find that, even with age structure, social learning can evolve without increasing population fitness, i.e., "Rogers' paradox" still holds. However, a process of "demographic filtering", together with cultural transmission, can generate cumulative improvements in adaptation levels. We further show that older age classes have higher proportions of adaptive behavior when the environment is stable and adaptive behavior is hard to acquire but important to survival. Through individual-based simulations comparing temporal and spatial variability in the environment, we find a "copy-the-old"-strategy only evolves when social learning is erroneous and the opposite "copy-the-young"-strategy can function as a compromise between individual and social information use. Our results reveal that age structure substantially changes how culture evolves and provide principled empirical expectations about age-biased social learning and the role of demography in cultural adaptation.

## 1 Introduction

When should we expect individuals to adopt the behavior and norms of older generations? In her book "Culture and Commitment", anthropologist Margaret Mead suggested a prominent role for the prevalent rate of social change [1]. When social change is slow, juveniles grow up in an environment that resembles that of their parents and older generations serve as valued models and authorities. In so called gerontocratic societies, which include the Ancient Greek city state of Sparta and contemporary East-African pastoralists such as the Kenyan *Samburu* [2, 3], social stratification is predominantly based on age classes and opinions of older

com/DominikDeffner/Age-structured-Social-Learning.

**Funding:** This work has been funded by the Max Planck Society. The funders had no role in study design, data collection and analysis, decision to publish, or preparation of the manuscript.

**Competing interests:** The authors have declared that no competing interests exist.

individuals tend to be highly valued. When societal change is more rapid, on the other hand, being old may predict being out of date as relevant conditions might have changed since older generations acquired their norms and behaviors. Starting in the 1960s, attitudes associated with older generations are often seen as backward and outdated in Western societies and young individuals regularly attend to their peers instead [1, 4, 5]. Which demographic and cultural factors could explain these opposing views on the value of information provided by older individuals?

Children across societies need to learn many essential skills to become competent adult members of their communities. The transmission and gradual modification of cultural information over generations is often considered key to human evolutionary success as it allows us to flexibly adjust to vastly different environments through locally adapted tools, beliefs and institutions [6–9]. Past theoretical work has investigated the conditions that favor the evolution of social learning and the way individuals should combine individual and social information strategically [10–12].

One starting point was Alan Rogers' equally simple and elegant model that demonstrates how pure social learning does not increase mean population fitness because its adaptive value is strongly frequency-dependent [13]. With few other social learners, chances are high to copy an individual learner, who has learned from direct interactions with the environment, providing adaptive information without costly trial-and-error learning. As the proportion of social learners increases, fewer and fewer individuals track the state of the environment, which can result in maladaptive information cascades [14]. At equilibrium, there tends to be a mix of individual and social learners that have the same mean fitness as a population entirely comprised of individual learners. This observation has become famous as "Rogers' paradox" in the literature—even though there is nothing inherently paradoxical about it, other than violating many researchers' intuitions.

This and most following models omit age structure and study the evolution of learning in populations with discrete, non-overlapping generations. Individuals are assumed to be born, learn from the previous generation, reproduce and then all die at the same time. Social learning and life history, however, interact in human adaptation and we need further modeling work that incorporates age structure and different population dynamics [15]. Age structure is a central feature of human and other animal populations, can have profound consequences for evolutionary dynamics [16], and is also expected to shape the informational environment learning is responding to. To address how culture facilitates flexible human adaptation and when selection favors learning from older generations, we need formal theory of the evolution of social learning in populations with age classes. The few previous models that did include age structure divided the population into a small number of fixed age categories and investigated the cultural and demographic consequences of transmission modes that varied across life stages (e.g., [17–19]) or solved for learning schedules that could support cumulative culture [20]. We still require a game-theoretic life-history theory of learning that does not externally impose age structure but has it emerge from basic assumptions about survival and reproduction and focuses on the frequency-dependent nature of the evolution of social learning.

The goal in this paper is to develop the simplest possible model of the classic adaptive value of culture, but in an age-structured population. So call this "Rogers model with age classes". As such, it isn't meant to represent any specific organism. Rather it represents the structure of an argument. How does demographic population structure and social learning interact in adaptation and population growth and what are the minimal conditions for such forces to make it worthwhile to copy older individuals? We first formulate a mathematical model of the population and learning dynamics and demonstrate that adding age does not resolve Rogers' paradox. The differential survival of adapted individuals to high ages (what we call "demographic

filtering"), however, can lead to higher frequencies of social learning and cumulative improvements in adaptation levels. We then solve for the conditions that favor learning from older vs. younger individuals. Finally, we confirm and extend the analytical results through individual-based simulations and investigate how age-biased social learning evolves in temporally and spatially variable environments.

## 2 The model

Consider a large population of perennial organisms that live in a temporally varying environment. The environment can take on one of a very large number of states. Each state has a corresponding adaptive response. An adaptive response may increase both survival and fertility of adults. The environment may change with probability $u$ each time step, rendering all previously adaptive behavior non-adaptive.

Individuals are born as *juveniles* who cannot yet reproduce, but may learn about the environment. After one time step, a juvenile who survives transitions to adulthood. In the analytical model, adults do not learn, but do reproduce (in the individual-based simulations, we later also allow for adult learning). This implements the extreme form of an exploration–exploitation trade-off between investment in learning and investment in reproduction that all organisms face. Focusing on pure mechanisms helps to reveal causal impacts. Let $s_j$ be the probability of survival of a juvenile or an adult who practices response $j$, where $j = 1$ indicates adaptive behavior and $j = 0$ indicates non-adaptive behavior. Let $b_j$ be the fertility of an adult with response $j$. Juveniles always have zero fertility. They must survive once to have any chance of producing descendants.

The organism's challenge is to find the currently adaptive response. Juveniles may learn either socially or individually. Social learners sample two models and then may choose among them based upon relative age. Individual learners instead reduce their probability of survival to adulthood by a factor $c$ for a chance $z$ of innovating an adaptive response.

### 2.1 Recursions

The state of the population is given by the state variables $n_{ij}$, where $i$ is an age class and $j = 1$ indicates an individual with adaptive response. The number of juveniles $n_{10}$ (all are born non-adapted) is regenerated each time period by fertility from all adult age classes:

$$n'_{11} = p\sum_{i=2}^{\infty}\sum_{j=0}^{1} n_{ij}b_j, \tag{1}$$

$$n'_{10} = (1-p)\sum_{i=2}^{\infty}\sum_{j=0}^{1} n_{ij}b_j, \tag{2}$$

where $p$ is the probability a juvenile acquires currently adaptive response through learning. We define $p$ as a function of heritable strategies in the next section. The number in each adult age class $i > 1$ is given instead by:

$$n'_{i0} = s_0 n_{i-1,0} + u s_1 n_{i-1,1}, \tag{3}$$

$$n'_{i1} = (1-u)s_1 n_{i-1,1}. \tag{4}$$

Since adults do not learn in this model, these recursions are rather simple. However, when the environment changes with probability $u$, all adapted adults are rendered forever non-

adapted. There is a special case for $i = 2$ that includes the cost of innovation $c$:

$$n'_{20} = (s_0 n_{1,0} + u s_1 n_{1,1})(1 - c\pi), \tag{5}$$

$$n'_{21} = ((1 - u) s_1 n_{1,1})(1 - c\pi). \tag{6}$$

The symbol $\pi$ gives the probability of individual learning and is further defined in the next section.

## 2.2 Learning

Let $p$ be the probability a juvenile acquires currently adaptive behavior. We define this as a function of a strategy vector $\{\pi, \phi\}$, where $\pi$ is the probability of individual learning and $\phi$ determines the direction and strength of any age bias for models to socially learn from. Specifically:

$$p(\pi, \phi) = \pi z + (1 - \pi) Q(\phi), \tag{7}$$

where $z$ is the probability an individual learner acquires adaptive behavior. $Q$ is a function for the probability of acquiring adaptive response by social learning. We define $\phi$ as the proportional odds of copying the older of two individuals. Let $R$ be the probability of copying the older of two individuals. This implies that the probability of copying the older individual is given by solving this expression for $R$:

$$\phi = \frac{R}{1 - R}, \tag{8}$$

yielding $R = \phi/(\phi + 1)$. To compute $Q$, we need to average over all possible pairs of social models. Let $q_i$ be the proportion of individuals in age class $i$ with adaptive behavior. Let $a_i$ be the proportion of adults ($i > 1$) in age class $i$. Then the probability a juvenile acquires adaptive behavior through social learning is:

$$Q(\phi) = \sum_{i=2}^{\infty} a_i^2 q_i + \sum_{i=3}^{\infty} \sum_{j=2}^{i-1} 2 a_i a_j \left( q_i \frac{\phi}{\phi + 1} + q_j \frac{1}{\phi + 1} \right). \tag{9}$$

In the above expression, the first summation is all pairs with tied ages. In these pairs, there is no age asymmetry. So the learner imitates at random and acquires adaptive response when a random individual of age class $i$ has an adaptive response. In the second summation, we sum over asymmetric pairs where an individual in age class $i$ is older than one in age class $j$. There is a probability $2 a_i a_j$ of such a pair, allowing for both orderings, and then the learner acquires adaptive response according to which model is imitated and whether an individual of that age class has an adaptive response. When $\phi = 1$, the above simplifies to random copying of any adult: $Q(1) = \sum_{i=2}^{\infty} a_i q_i$.

## 2.3 Individual-based simulations

To corroborate the analytical predictions and extend the model to more complex scenarios that also account for adult learning and different sources of stochasticity in finite populations, we construct an individual-based version of the model with either temporally or spatially varying environments.

**Temporal variation.** In the temporal model, we consider a population of fixed size with $N = 1000$ individuals. After birth, each juvenile engages in either individual or social learning according to a dichotomous genotype. Individual learners acquire adaptive behavior with

probability $z$ and pay a recruitment cost $c$. Social learners randomly sample two interaction partners from the adult population and copy the behavior of either the older or younger individual depending on an inherited learning strategy. If both interaction partners have the same age, they choose among them at random. Individuals acquire adaptive behavior from an adapted interaction partner with probability $1 - \epsilon$, where $\epsilon$ represents the degree of copying error. All individuals survive to the next age class with probability $s_0$ (non-adapted) or $s_1$ (adapted). We assume asexual, haploid reproduction and all adults have an equal opportunity to give birth to offspring to fill the empty spots in the population. Juveniles inherit the genotype, which determines learning strategies, from their parent with a small probability, $\mu =$ 0.005, of mutation for both loci. Each time step, there is a probability $u$ that the environment changes. When environmental change occurs, all variants in the population become non-adaptive.

**Spatial variation.** For the spatial model, we consider a metapopulation of fixed size with $N = 2000$ individuals. The metapopulation is divided into 4 equal sub-populations linked by migration. Life cycle and learning process are identical to the temporal model. The only exception is that, instead of temporal changes in the environment, there is now constant migration between different habitats. Habitats are assumed to differ in important environmental parameters such that each habitat is characterized by a different adaptive response. Specifically, per time step, each adult has a probability $m$ of migrating to another habitat which results in a loss of adaptive behavior. Note that the way migration is implemented ensures that population sizes in all sub-populations stay constant over time.

## 3 Analysis

### 3.1 Does adding age structure resolve Rogers' paradox?

As a first step, we investigate whether age structure resolves "Rogers' paradox", i.e., the finding that social learning can invade a population of individual learners without increasing mean population fitness. We compare a baseline model with discrete, non-overlapping generations (i.e., the original Rogers model) to the age-structured learning model described above. For this analysis, as in the Rogers model, we assume individual learning always produces adaptive behavior (i.e., $z = 1$) and social learners copy adults randomly (i.e., $\phi = 1$). Relaxing these assumptions produces the same qualitative results. Fig 1 shows proportions of social learners (top) and fitness (bottom) for the original Rogers model (left) and the age-structured version (right). Fitness is calculated as the lineage growth rate at equilibrium $\lambda$ which is the correct measure to use in age-structured populations [16]. Unsurprisingly, in both models, social learning evolves more readily as individual learning is more costly and the environment changes less frequently (Fig 1A and 1B).

Fig 1C and 1D show fitness for populations with only individual learners (black shapes) and populations with individual and social learners (colored lines). This comparison reveals that fitness in both models is solely determined by the fitness of individual learners and does not change with the introduction of social learners. Age structure, thus, does not resolve Rogers' paradox. As a direct consequence of age structure and the assumption that only juveniles learn, fitness in the age-structured model decreases as the environment becomes more unstable. With discrete generations, the whole population gets replaced each time step, so fitness of individual learners does not depend on environmental stability. With age structure, however, individual learners can lose adaptive behavior during their lifetime through environmental change resulting in the decline in population fitness.

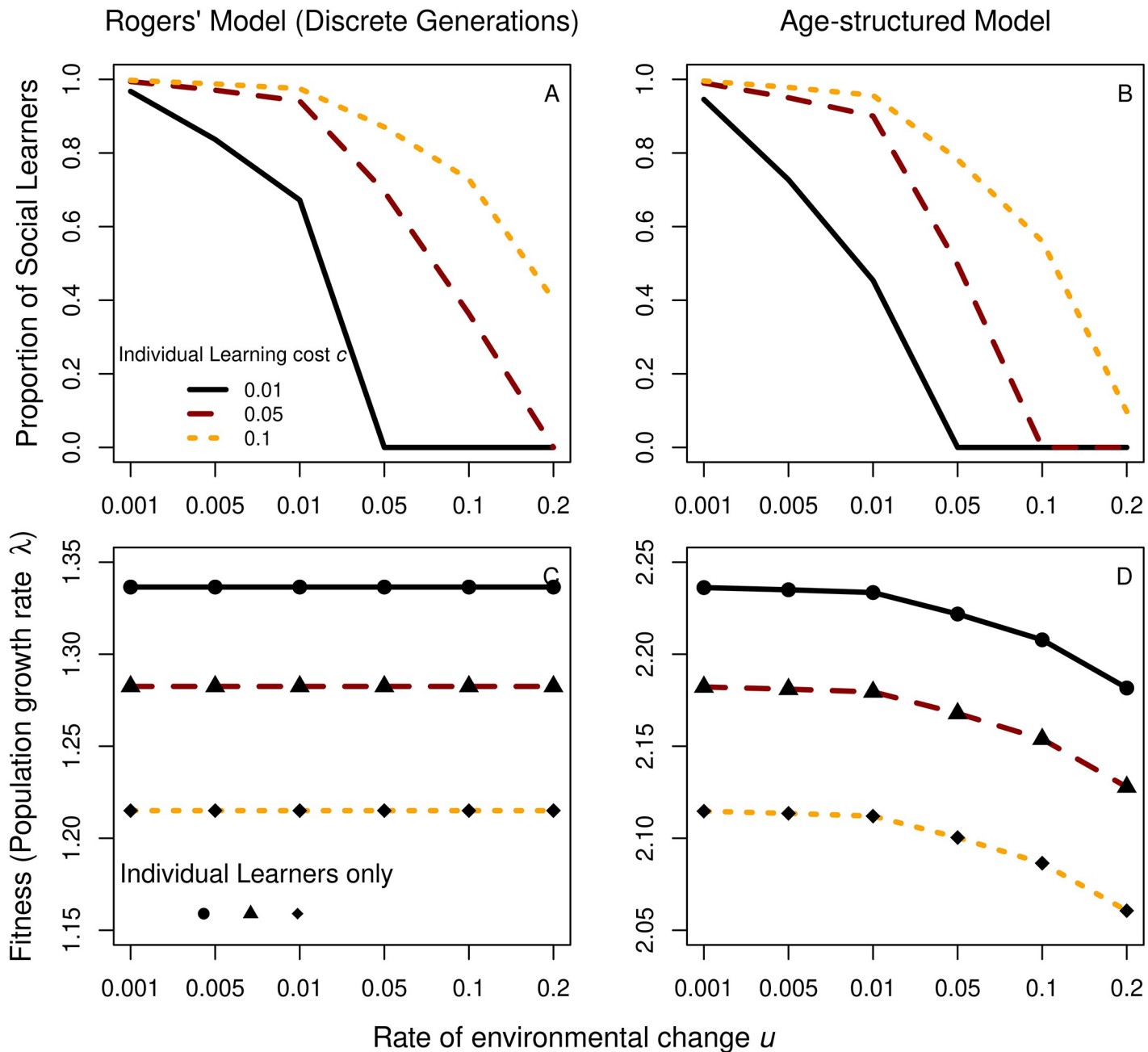

**Fig 1.** Proportion of social learners (top) and mean population fitness (lineage growth rate at equilibrium λ; bottom) conditional on the rate of environmental change $u$ and the cost of individual learning $c$. Results for the original Rogers model with discrete, non-overlapping generations are shown on the left, results for the age-structured model with overlapping generations on the right. Lines in plots C and D represent populations with social learners, black shapes give corresponding values for populations with only individual learners. Other parameter values were: $z = 1$, $\phi = 1$, $b_0 = 1$, $b_1 = 1.5$, $s_0 = 0.9$, $s_1 = 0.9$.

## 3.2 Demographic filtering and its interplay with social learning

Next, we investigate the interplay between social learning and demography and their joint effect on the distribution of adaptive information across age classes and adaptation after a change in the environment. If there are differential chances of survival for adapted and non-adapted individuals (i.e., viability selection), the "demographic filtering" of adaptive behavior

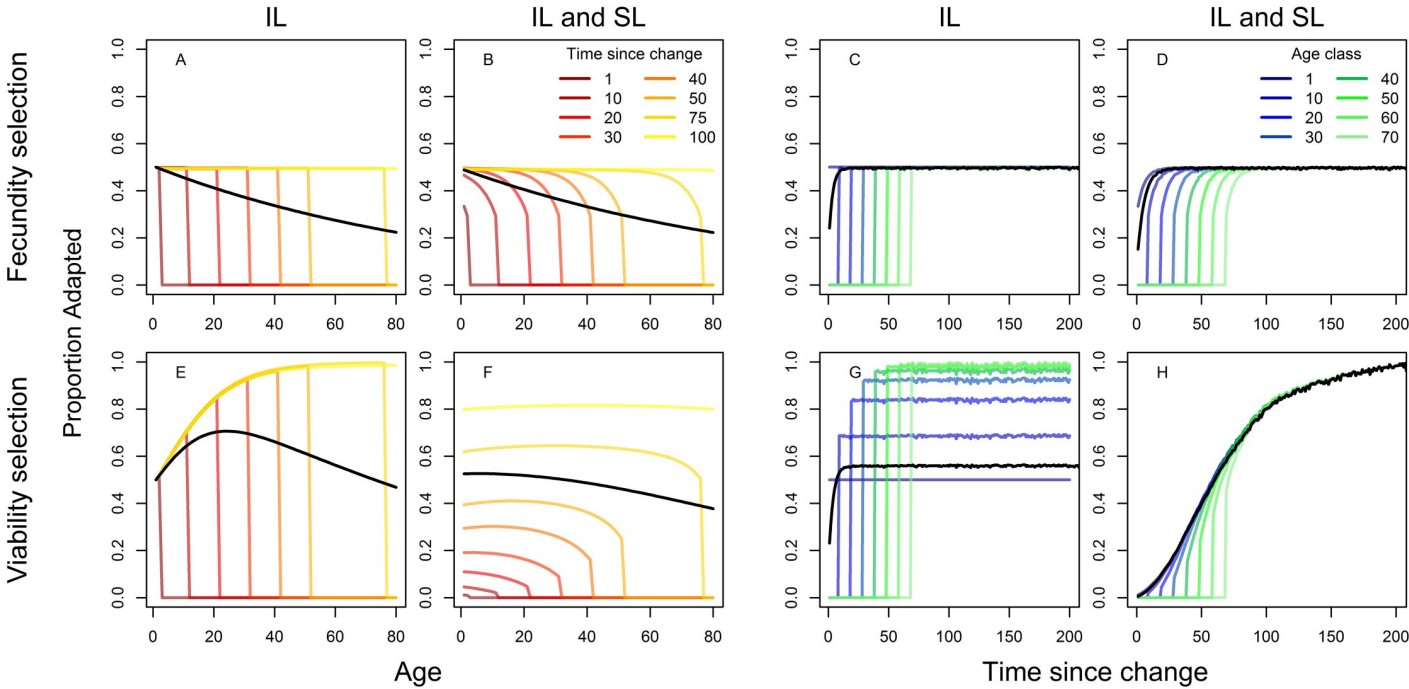

**Fig 2.** Proportion of adapted individuals per age class (left) and for different time intervals after an environmental change (right). Dynamics under fecundity selection (i.e., without demographic filtering) are shown on the top, dynamics under viability selection (i.e., with demographic filtering) are shown on the bottom. Comparison of populations with only individual learners (IL) and populations comprising both individual and social learners (IL and SL). Black solid lines represent averages. Results are shown for a $z = 0.5$ success rate of individual learning. Other parameter values were chosen to keep fitness constant at $\lambda \approx 1.2$ across different scenarios: $u = 0.01$, $c = 0.01$, $\phi = 1$, $b_0 = 0.35$, $b_1 = 0.5$ (for fecundity selection), $s_0 = 0.85$, $s_1 = 0.93$ (for viability selection).

in older age classes might constitute another adaptive force that social learners could benefit from. To illustrate the pure effects of this demographic filtering and cultural transmission, parameter values were chosen to keep fitness constant at $\lambda \approx 1.2$ across all comparisons.

**Individual learning only.** We start with a population of only individual learners and compare the behavior of the model with a stochastically changing environment under pure fecundity and pure viability selection. Fig 2A and 2E illustrate the effect of demographic filtering on the distribution of adaptive behavior across age classes. If there is no survival advantage to adaptive behavior (Fig 2A), older age classes are always expected to have lower proportions of adaptive behavior as environmental change periodically renders their behavior out of date. This effect is illustrated by the "cliffs" in Fig 2, where adaptive behavior suddenly drops to 0 above certain ages. If being adapted confers survival benefits (Fig 2E), average proportions of adaptive behavior increase up to a certain age as those possessing adaptive behavior will be more likely to survive to old ages. For populations of pure individual learners, we can calculate the proportion of individuals in age class $i$ with adaptive behavior, $q_{IL,i}$, and evaluate how the forces of demographic filtering and environmental change trade off:

$$q_{IL,i} = (1 - u)^{i-1} \frac{s_1^{i-1} z}{s_1^{i-1} z + s_0^{i-1} (1-z)}. \tag{10}$$

A first prerequisite is that the environment has not changed since an individual has learned, which is given by the first part of the equation. This term is multiplied by the relative

probability that adults have invented adaptive behavior as juveniles and survived up to age $i$. S1 Fig plots resulting age curves for different values of $u$, $\sigma = s_0/s_1$, and $z$. Expectedly, more rapid changes in the environment result in lower values and earlier peaks of adaptive behavior. Stronger viability selection leads to earlier peaks, as even surviving to relatively young ages indicates being adapted when selection acts strongly on chances of survival.

We now turn to levels of population adaptation after a change in the environment (Fig 2C and 2G). Resulting from demographic filtering, which leads to high adaptation levels in old age classes, populations under viability selection reach slightly higher average adaptation levels (Fig 2G) compared to populations under fecundity selection (Fig 2C). However, they still plateau at a value determined by the success rate of individual learning (here $z = 0.5$) because the benefits of adaptive filtering are limited to older age classes and cannot improve juvenile learning.

**Individual and social learning.** Now we also let social learners evolve. Demographic filtering increases the adaptive value of culture and leads to substantially higher frequencies of social learners in the population (for this parameter combination, means of 98% and 38% social learners for viability and fecundity selection, respectively). By increasing proportions of adaptive behavior in older age classes, demographic filtering improves the quality of social information in the population; this results in more copying among juveniles. As social learning effectively drains adaptive information accumulated through demographic filtering, it lowers the peak in adaptive behavior in older age classes (Fig 2F).

After an environmental shock, populations under fecundity selection (Fig 2D) quickly recover as they maintain high rates of individual learning. However, they are limited to the adaptation levels individual learners can obtain because innovation is the only adaptive force in the system. Under viability selection (Fig 2H), populations take longer to recover because few individual learners track the state of the environment. However, in the long run, adaptation levels exceed those of individual learners (Fig 2C and 2G) and of cultural populations without demographic filtering (Fig 2D) and increase until the whole population is adapted. Combining demographic filtering and social learning, the accumulated benefits of adaptive filtering can feed back on juveniles and provide them with high-quality social information. This process results in an adaptive interplay between cultural and demographic forces that is characterized by high proportions of adaptive behavior across age classes and much higher overall adaptation levels than acultural populations could obtain.

## 3.3 When are older individuals better adapted?

After considering the general impacts of age structure, we now turn to our starting question: When will selection favor a positive ($\phi > 1$) or negative ($\phi < 1$) age bias? To address this question, we require expressions for the $q_i$ state variables, the proportion of each age class $i$ with adaptive behavior. Specifically, we would like to know, under a separation of genetic and demographic time scales (see chapter 5.4. in [21]), the demographic environment that selection of $\phi$ responds to. This means we seek expressions for the steady state $\hat{q}_i$ values corresponding to a stable age distribution.

Since this model contains no population regulation (see [15] for effect of different forms of population regulation on social learning), the population will either increase to infinity or decline to zero. We are interested in the former case. In that case, the frequency of individuals in each age-and-behavior class $ij$ will eventually stabilize, even though the number of individuals in the population will continue to grow. Therefore we analyze the frequencies $\hat{f}_{ij} = n_{ij}/N$ that define the stable age-and-behavior distribution. These frequencies are defined by the recurrence equations in section 2, normalized by $N$. This normalization introduces a

population growth adjustment. To see this, consider the simplest case:

$$\hat{f}_{i,1} = \frac{n'_{i,1}}{N'} = \frac{1}{N'}(1-u)s_1\, n_{i-1,1}.$$ (11)

Since we consider now the stable age-class distribution, we divide both sides by $N$ so we relate stable frequencies:

$$\frac{\hat{f}_{i,1}}{N} = \frac{1}{N'}(1-u)s_1\, \frac{n_{i-1,1}}{N}$$ (12)

$$= \frac{1}{N'}(1-u)s_1\hat{f}_{i-1,1}.$$ (13)

Finally, move the $N^{-1}$ on the left to the right and note that $N'/N = \lambda$, the stable population growth rate:

$$\hat{f}_{i,1} = \frac{N}{N'}(1-u)s_1\hat{f}_{i-1,1} = \frac{1}{\lambda}(1-u)s_1\hat{f}_{i-1,1}.$$ (14)

The same approach yields the recurrence for non-adapted individuals at age $i$:

$$\hat{f}_{i,0} = \frac{1}{\lambda}\left(s_0\hat{f}_{i-1,0} + us_1\hat{f}_{i-1,1}\right).$$ (15)

Note that when the population grows slowly, $\lambda \approx 1$. This is a common assumption in life history analysis [21, 22], but we will attempt to keep $\lambda$ general, in case further insights arise from the generality. These equations can be solved explicitly, yielding formulas in terms of only initial conditions and parameters (see analysis notebook in GitHub repository). For $i > 1$:

$$\hat{f}_{i,0} = \hat{f}_{1,0}\left(\frac{s_0}{\lambda}\right)^{i-1} + \hat{f}_{1,1}us_1\frac{\left(\frac{s_0}{\lambda}\right)^{i-1} - s_1^{i-1}(1-u)^{i-1}}{s_0 - \lambda s_1(1-u)},$$ (16)

$$\hat{f}_{i,1} = (\lambda^{-1}(1-u)s_1)^{i-1}\hat{f}_{1,1}.$$ (17)

Remember that we are looking to identify the stable proportion of individuals at age $i$ with adaptive behavior. This is defined as:

$$\hat{q}_i = \frac{\hat{f}_{i,1}}{\hat{f}_{i,1} + \hat{f}_{i,0}}.$$ (18)

This expression depends implicitly upon $\hat{p}$, the stable probability that a juvenile acquires adaptive behavior. But $\hat{p}$ depends in turn upon the $\hat{q}_i$ values. Fortunately, we do not need to determine $\hat{p}$ at steady state, we just need to use it implicitly in our solution. Specifically, note that the expressions for $n'_{1,1}$ and $n'_{1,0}$ in section 2 imply:

$$\hat{p} = \frac{\hat{f}_{1,1}}{\hat{f}_{1,1} + \hat{f}_{1,0}}.$$ (19)

Equivalently:

$$\frac{\hat{p}}{1 - \hat{p}} = \frac{\hat{f}_{1,1}}{\hat{f}_{1,0}}. \tag{20}$$

This allows us to substitute $\hat{f}_{1,1} = \hat{f}_{1,0}\hat{p}/(1 - \hat{p})$, yielding (for $\lambda \approx 1$):

$$\hat{q}_i = \frac{\hat{p}s_0 s_1^i (s_0 - s_1(1 - u))(1 - u)^i}{s_0^i s_1 (1 - u)(us_1 - (1 - \hat{p})(s_1 - s_0)) - \hat{p}s_0(s_1 - s_0)s_1^i(1 - u)^i}. \tag{21}$$

The expression for general $\lambda$ is complicated and rather non-transparent (see analysis notebook in GitHub repository). Now we can ask when $\hat{q}_i > \hat{q}_{i-1}$, i.e., when older age classes are expected to have higher proportions of adaptive information, yielding the condition (again for $\lambda \approx 1$):

$$\hat{p} < 1 - u\frac{s_1}{s_1 - s_0}. \tag{22}$$

Or equivalently, letting $\sigma = s_0/s_1$:

$$\hat{p} < 1 - \frac{u}{1 - \sigma}. \tag{23}$$

This is the condition for older age classes to have larger proportions of adaptive behavior and therefore the precondition for selection to favor $\phi > 1$. Neither copying the old nor the young is always beneficial. Rather, whether or not older individuals are better adapted depends upon a balance of forces. The force of environmental change $u$ is to reduce the proportion of adaptive behavior in older age classes—as $u$ increases, the condition above becomes increasingly difficult to satisfy. The force of a survival advantage to adaptive behavior, in contrast, is to increase adaptive behavior in older age classes. As $\sigma$ decreases, corresponding to a larger advantage to adaptive behavior, the condition is easier to satisfy.

This condition confirms the intuition behind the model and also provides quantitative guidance on the relative strength of these forces. In Fig 3, we plot the threshold value of $\hat{p}$ across all possible values of $\sigma$, for different values of $u$. The region below each curve corresponds to combinations of $\hat{p}$ and $\sigma$ that lead to older age classes having higher proportions of adaptive behavior. When environments change very rapidly, the adaptive behavior must be hard to acquire for a juvenile and must provide large survival benefits for older individuals to be better adapted. On the other hand, when environments are very stable, the above condition is almost always fulfilled, such that older individuals are better adapted irrespective of other factors.

## 3.4 The evolution of age-biased social learning in temporally and spatially varying environments

According to Eq 23, juveniles should be more likely to copy the older of two individual if the environment is relatively stable, adaptive behavior is hard to acquire and there is a large survival advantage to adaptive behavior. Which of these forces is strongest depends not only on their parameters, $\sigma = s_0/s_1$ and $u$, but also on the frequency of adaptive behavior in the population. To confirm and extend the analytical results, we analyze the stochastic individual-based version of the model and explore how readily age-biased social learning actually evolves in both temporally and spatially varying environments.

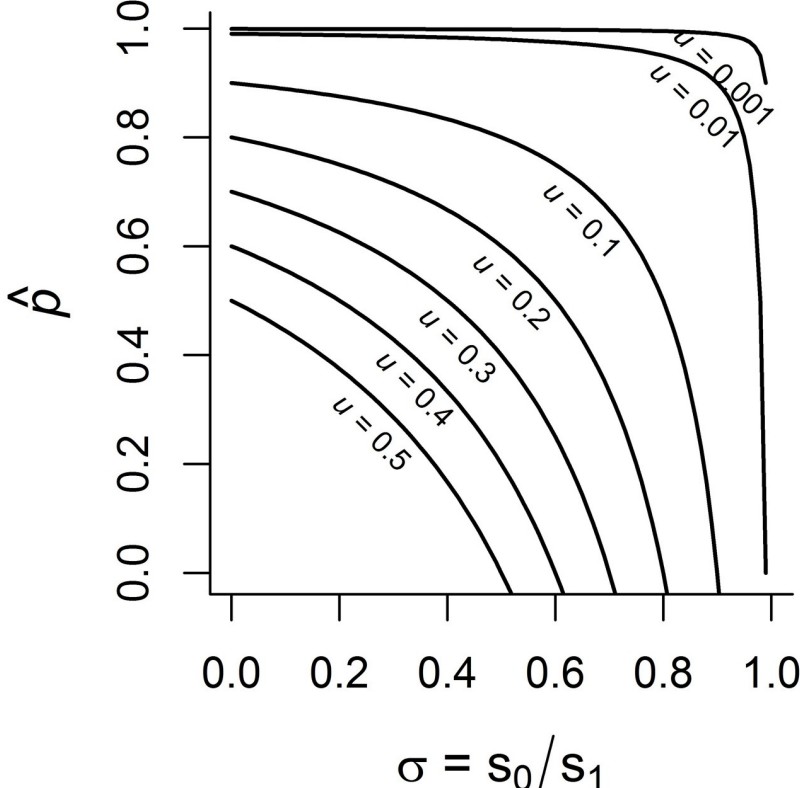

**Fig 3. Threshold values as defined by expression 23 for different rates of environmental change $u$.** Areas underneath each curve indicate parameter combinations of $\hat{p}$ (probability to acquire adaptive behavior) and $\sigma$ (strength of viability selection) for which older age classes are expected to have larger proportions of adaptive behavior.

Fig 4 shows proportions of social learners (top) and of individuals biasing their learning towards older vs. younger individuals ("Old Bias"; bottom) for both the temporal (left) and spatial model (right). Lighter colors indicate higher proportions. In the temporal model, social learning evolves when the environment is relatively stable or when there is no or very little advantage to possessing adaptive behavior ($\sigma \approx 1$). Notably, there are no regions in parameter space that clearly favor copying the old. Instead, when the environment changes frequently, but not rapidly enough to favor large numbers of individual learners, evolution favors a pronounced "copy-the-young" bias.

In the spatial model, high proportions of social learning can also occur with much more unstable environments (i.e., high migration rates) in simulations with strong viability selection. This is because, under such circumstances, selection continuously removes non-adaptive variation brought in by migrants, such that most group members possess adaptive behavior and copying is advantageous. Here, selection favors a slight "copy-the-old" bias in regions with stable environments and/or large advantages to adaptive behavior, as predicted by the analytical model.

While $u$ and $\sigma$ are parameters that we can directly manipulate, $\hat{p}$, the third variable in Eq 23, arises endogenously from different interacting forces within the model. The reason no stronger "copy-the-old" bias can evolve in either model, is that in stable environments with large benefits to adaptive behavior—conditions that would favor old bias—essentially everyone will be adapted. As a consequence, any social learning strategy will result in the acquisition of

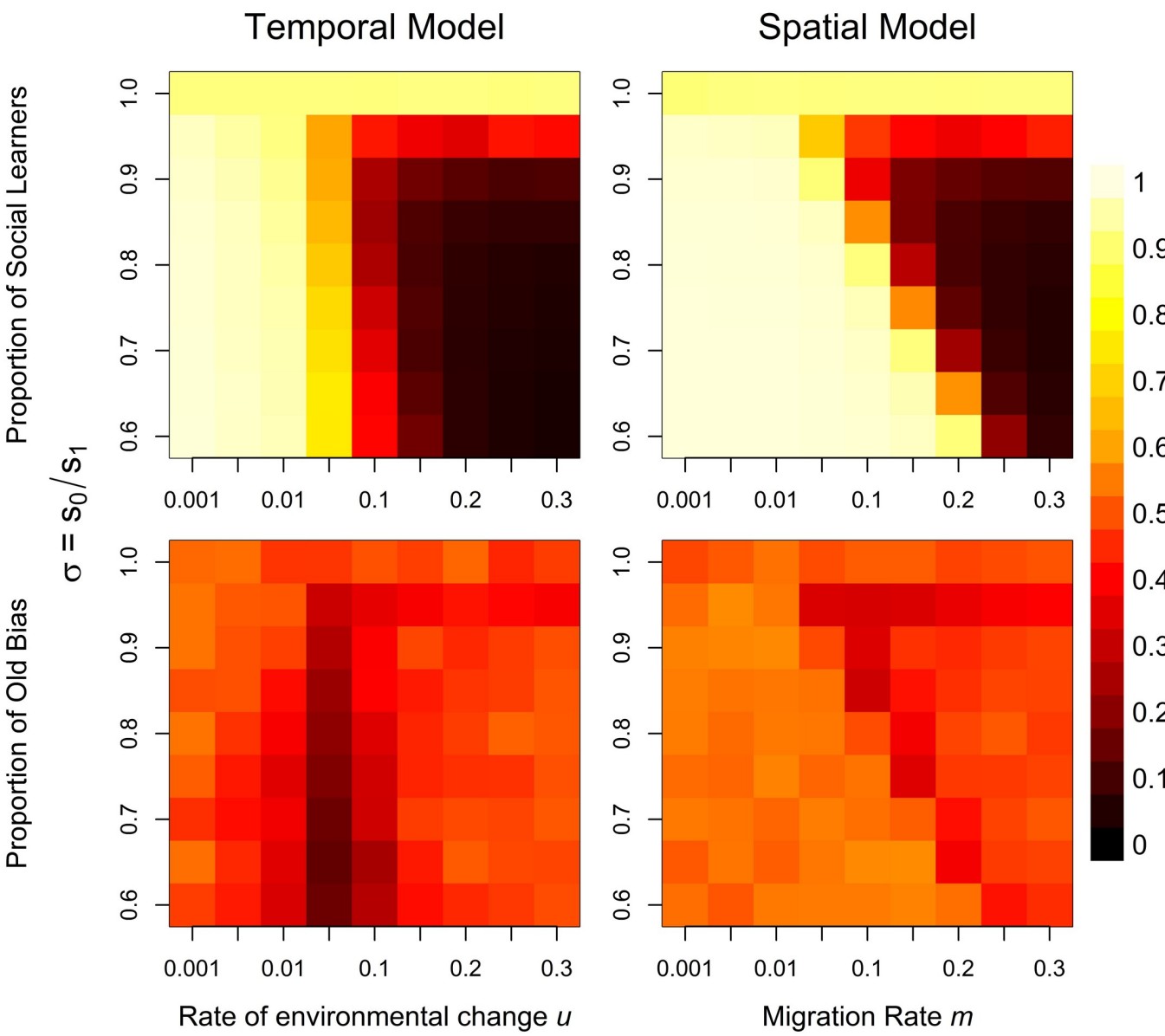

**Fig 4.** Proportion of social learners (top) and old bias (bottom) for the temporal (left) and spatial (right) individual-based model. Squares represent different parameter combinations for the rate of environmental change $u$ / migration rate $m$ and the strength of viability selection $\sigma$. Lighter colors indicate higher proportions, darker colors lower proportions. Results for both models are averaged over the last 5000 time steps of 10 independent 7000 time-step simulations per parameter combination. Other parameter values are: $c = 0.05$, $z = 0.5$, $s_1 = 0.9$, $\mu = 0.005$.

adaptive behavior and there is a strong ceiling effect limiting the scope of demographic filtering. So far, we have assumed there is no copying error, $\epsilon = 0$, i.e., social learners can perfectly reproduce the behavior they observe in a model. This is a rather strong assumption and in reality there might be many circumstances where juveniles fail to acquire a trait through social learning [23]. S2 Fig shows results for simulations where social learning results in the successful adoption of adaptive behavior in only 70% of learning events (i.e., $\epsilon = 0.3$). By lowering $\hat{p}$, this slight modification leads to the evolution of large proportions of "copy-the-old" bias in both the temporal and spatial model. When social learning is imperfect, there is room for

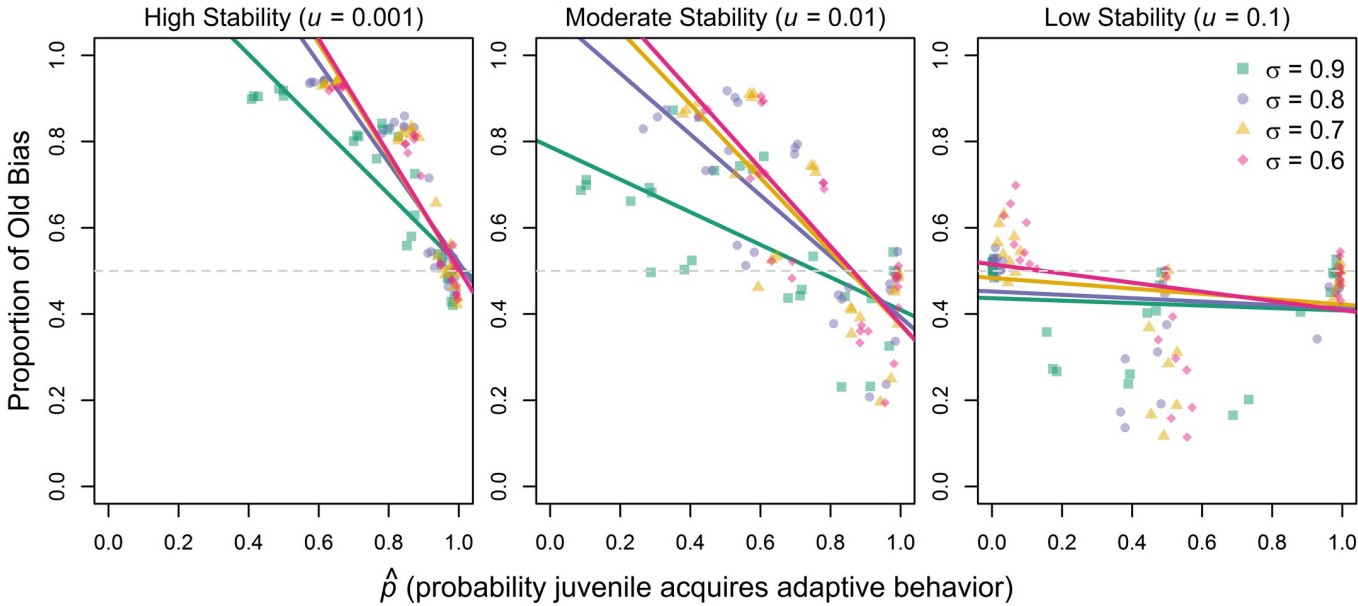

**Fig 5.** Proportions of old bias conditional on $\hat{p}$, the probability juveniles acquire adaptive behavior, for high (left), moderate (middle) and low (right) environmental stability. Results are taken from temporal individual-based model. Each point represents mean across last 5000 timesteps of 10 independent 7000 timestep simulations per parameter combination. Colors/shapes indicate different values of $\sigma$, the survival advantage of adaptive behavior. In addition to $u$ and $\sigma$, we also varied the cost of individual learning ($c = 0.05$, $c = 0.1$, $c = 0.2$), the success rate of individual learning ($z = 0.01$, $z = 0.5$, $z = 1$) and the error rate of social learning ($\epsilon = 0$, $\epsilon = 0.1$, $\epsilon = 0.3$), resulting in 972 parameter combinations. Lines (best linear fits) serve purely illustrative purposes.

demographic filtering to increase proportions of adaptive behavior in older age classes which turns older individuals into preferable models. This effect is also illustrated in Fig 5 which plots the proportion of old bias against $\hat{p}$ as calculated from simulation results. These results confirm that individuals are less likely to copy older individuals if juveniles have high chances to acquire adaptive behavior. This relationship holds for different combinations of $u$ and $\sigma$.

Finally, to better understand the influence of the assumption that only juveniles learn and to determine the generality of our findings, we run additional individual-based simulations with temporally changing environments and now also let adults update their behavior with a certain age-dependent probability. Specifically, we assume that the probability to learn (through either individual or social learning) declines exponentially, with parameter $\beta$ giving the rate of age-related decline: $e^{-\beta(Age-1)}$. This formulation implies that juveniles ($Age = 1$) always learn and adults become less likely to update their behavior as they become older. Fig 6 shows proportions of social learners (left) and old bias (right) for different age-related exponential decline rates in learning. If juveniles are substantially more likely to learn compared to older individuals (top rows), the results replicate previous findings where only juveniles could learn. As more older adults learn each time step and, thus, are less likely to become outdated, social learning becomes more prevalent in the population and learners progressively shift towards older versus younger role models. However, even if all adults learn each time step (bottom row), there is only a small region in parameter space—marked by moderate environmental stability and strong viability selection—that favors a pronounced "copy-the-old" bias even though older individuals have had more learning opportunities. Moreover, S3 Fig reveals that adult learning does not qualitatively change results for analyses with social learning errors. Overall, these analyses show that our results do not depend on a specific assumption about

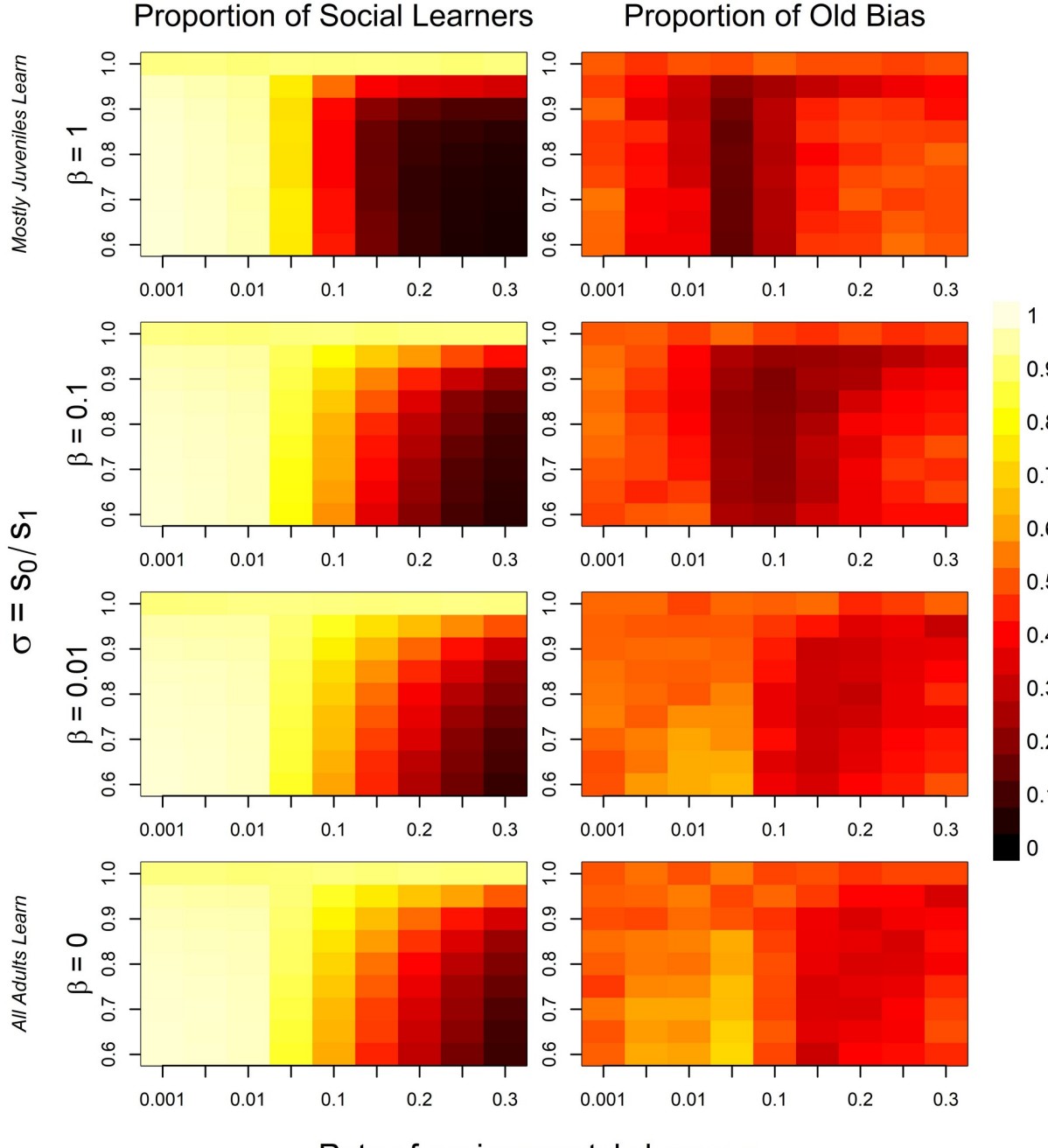

**Fig 6.** Proportion of social learners (left) and old bias (right) for temporal individual-based model with adult learning. Rows show results for different age-related exponential decline rates in learning; $\beta = 1$ means mostly juveniles learn, $\beta = 0$ means all age classes are equally able to learn. Squares represent different parameter combinations for the rate of environmental change $u$ and the strength of viability selection $\sigma$. Lighter colors indicate higher proportions, darker colors lower proportions. Results are averaged over the last 5000 time steps of 10 independent 7000 time-step simulations per parameter combination. Other parameter values are: $c = 0.05$, $z = 0.5$, $s_1 = 0.9$, $\mu = 0.005$.

exclusively juvenile learning, but on the biologically plausible assumption that adults become less likely to change their behavior as they age.

## 4 Discussion

Culture and demography jointly facilitate flexible human adaptation, yet most previous models decided to leave out demographic complexity and studied the evolution of learning and cultural dynamics in isolation. Making simplifying assumptions is of course important but even relatively basic questions, such as "When does selection favor learning from the old?", cannot be answered or even formally asked unless age structure is taken into account. Here, we have developed and analyzed the age-structured version of the classic Rogers model of the adaptive value of culture. As such, our model also omits a good deal of real-world complexity. The goal of theoretical modeling is not to approximate the natural world as closely as possible but to isolate the fundamental structure of an overly complex reality and apply formal logic to deduce basic causal forces [21, 24–26].

First, we found that adding age structure does not resolve Rogers' paradox. In populations with age classes, pure social learning still does not increase mean population fitness compared to individual learners alone. Theory has shown that for culture to increase population fitness, it must make individual learning either more accurate or less costly [27]. Enquist and colleagues [28], for instance, showed that a strategy of "critical social learning", where individuals switch to individual learning if social learning proves unsatisfactory, outcompetes pure individual and social learning strategies (see [29] for similar results investigating a "hybrid" learning strategy). In the present model, age structure can increase the adaptive value of culture, such that a population can maintain higher proportions of copying, but social learning cannot increase the fitness of individual learners. As both types of learners must per definition have the same fitness at equilibrium, social learning cannot increase mean population fitness. In a stochastic model with spatial population structure, Rendell and colleagues [30] found that, under some circumstances, pure social learning can increase average fitness. They also show that spatial structure can introduce a new paradox with social learning spreading even when it decreases average fitness. Individuals in this model occupied different cells within a square toroidal environment with both dispersal and learning occurring locally. This creates an "edge effect" where social learners can spread because they have greater fitness in contact zones between genotypes (due to higher chances to copy individual learners), even though their average fitness is lower. Future work should study the effects of different kinds of population structure and their interaction with each other and cultural dynamics.

Even though social learning did not increase mean population fitness, in combination with demographic filtering, it enabled the population to reach much higher proportions of adapted individuals. Demographic filtering increases proportions of adaptive behavior in older age classes and thus constitutes a second adaptive force in addition to individual learning. While this source of adaptive information is unavailable to populations of pure individual learners, social learners can utilize it by copying older generations. Thereby, social learners can spread to higher frequencies and simultaneously increase adaptation levels across age classes. Previous work has investigated how social learning and culture help organisms adapt to changing environments, but cultural adaptation is intimately related to the way organisms grow, reproduce and die [31, 32]. Humans show highly developed abilities to learn from others, but we also exhibit a prolonged childhood and juvenile period, shorter intervals between births, and a significant post-reproductive lifespan. To explain how these conspicuous features might have coevolved with our unique reliance on culture, we need more modeling work on the adaptive interplay between social learning and different life history dynamics [15]. As steps in this

direction, our results demonstrate that more realistic demographies can produce unique preconditions for social learning to flourish, and cumulative cultural improvement can arise from purely demographic processes in a model without built-in cumulative cultural evolution.

Our starting question was when natural selection would favor learning from older vs. younger individuals. This question has attracted considerable empirical attention (see [33] for a review on model-based biases in social learning), yet we still lack a principled theoretical framework to unify existing findings and generate new predictions. In developmental psychology, 15-month-olds were reported to be more likely to copy behaviors performed by an adult versus a two-year-old child [34] and three- and four-year-olds preferentially attended to information provided by an adult over a child in a object labelling task [35]. Studies also showed that children faithfully copy both relevant and irrelevant actions demonstrated by adult models but only relevant actions demonstrated by children [36, 37]. Research in anthropology suggests that hunter–gatherer social learning is primarily vertical under age 5 and oblique and horizontal between the ages of 6 and 12 [38]. Finally, adult participants in a collective learning experiment, which implemented population turnover through a schedule of migration and environmental change, also preferentially copied more experienced (or "older") group members [39]. Taken together, these studies seem to imply that under most circumstances learners preferentially attend to older, more experienced, individuals. Our modeling results show that this can be a good strategy, especially when the environment is relatively stable, adaptive behavior is hard to acquire and confers large survival advantages. However, we note that any agreement with empirical results could of course also be due to missing factors. Many essential skills in real animals, and humans in particular, take generations to evolve and years to develop, which is particularly true for cumulative and causally opaque human culture [15, 40]. If adaptive behaviour takes time and practice to develop, older individuals may be more likely to have mastered high skill levels and may therefore—not because of demographic filtering—be preferable cultural role models.

Moreover, we found through simulation that due to ceiling effects in adaptive behavior a "copy older over younger models" strategy can only reliably evolve when social learners occasionally fail to copy adaptive behavior, even if adults can repeatedly update their behavior. The opposite "copy younger over older models" strategy can be advantageous when the environment fluctuates frequently but still maintains large proportions of social learners. This means, organisms might use a "copy-the-young" strategy as a compromise between individual and social information use when only young individuals are likely to have updated their behavior since the last change in the environment. Similar to our findings, the best strategies in the second round of the social learning strategies tournament copied both successful and young demonstrators [41, 42]. Individuals in their simulations could update behavior throughout their lifetime, so one might expect older individuals would be better adapted due to more learning opportunities. At each point in time, individuals could either exploit known behaviors to obtain their rewards or learn a new behavior. As a consequence, individuals mostly learned right after birth or after a change in the environment when they experienced a drop in payoffs. Because births were random with respect to changes in the environment, juveniles were more likely to acquire adaptive behavior after birth compared to adults who mostly learn when the environment has just changed. In line with our simulation results including adult learning, this shows that selection can favor learning from the young as long as there is some exploration–exploitation trade-off between investment in learning and investment in reproduction and not only when learning is restricted to pre-reproductive juveniles, the extreme form of this trade-off.

Our model contributes to a greater theoretical understanding of how culture helps organisms adapt, but it also generates several empirical predictions on age-biased social learning

that researchers could start testing in both humans and other animals. For example, researchers could identify which behaviors in a given population are most strongly associated with longevity and investigate if there are differences in who juveniles learn those behaviors from. The model predicts that age should more heavily be used as a cue for behaviors that improve survival. Similarly, researchers could investigate how difficult it is to innovate a behavior or to adopt it though social learning and determine whether learners rely more on age cues for behaviors that are particularly hard to acquire (e.g., causally opaque and/or normative social information). Finally, our model provides predictions about the influence of rates of environmental change: Learners should rely more on older role models to learn behaviors for which relevant environmental parameters are relatively stable and focus on younger models, instead, when parameters change frequently. In general, predictions are expected to apply both to differences in relevant parameters among populations and to different behavioral contexts within a population.

Summarizing, in this paper, we developed and analyzed an age-structured model of the evolution of social learning. Even in such simple models, we find intricate interactions between culture and demography that change our understanding of how cultural organisms learn and adapt. We are just beginning to understand how such joint culture-demography-systems might behave in general and further bodies of theory including strategic learning and cumulative culture are necessary to untangle the co-evolutionary relationships between demography and culture.

## Supporting information

**S1 Fig. Proportions of adapted individuals per age class in population of individual learners, as defined by expression 10 in the main text.** Plots show results for different values of $z$ (top row: $z = 0.7$, bottom row: $z = 0.1$), $\sigma$ (left: $\sigma = 1$, center: $\sigma = 0.75$, right: $\sigma = 0.5$) and $u$ (solid: $u = 0.001$, dashed: $u = 0.01$, dotted: $u = 0.1$.).
(TIFF)

**S2 Fig.** Proportion of social learners (top) and old bias (bottom) for the temporal (left) and spatial (right) individual-based model with social learning error ($\epsilon = 0.3$). Squares represent different parameter combinations for the rate of environmental change $u$ and the strength of viability selection $\sigma$. Lighter colors indicate higher proportions, darker colors lower proportions. Results for both models are averaged over the last 5000 time steps of 10 independent 7000 time-step simulations per parameter combination. Other parameter values are: $c = 0.05$, $z = 0.5$, $s_1 = 0.9$, $\mu = 0.005$.
(TIFF)

**S3 Fig.** Proportion of social learners (left) and old bias (right) for temporal individual-based model with adult learning and social learning error ($\epsilon = 0.3$). Rows show results for different age-related exponential decline rates in learning; $\beta = 1$ means mostly juveniles learn, $\beta = 0$ means all age classes are equally able to learn. Squares represent different parameter combinations for the rate of environmental change $u$ and the strength of viability selection $\sigma$. Lighter colors indicate higher proportions, darker colors lower proportions. Results are averaged over the last 5000 time steps of 10 independent 7000 time-step simulations per parameter combination. Other parameter values are: $c = 0.05$, $z = 0.5$, $s_1 = 0.9$, $\mu = 0.005$.
(TIFF)

## Author Contributions

**Conceptualization:** Dominik Deffner, Richard McElreath.

**Formal analysis:** Dominik Deffner, Richard McElreath.

**Funding acquisition:** Richard McElreath.

**Methodology:** Dominik Deffner, Richard McElreath.

**Software:** Dominik Deffner.

**Supervision:** Richard McElreath.

**Validation:** Dominik Deffner.

**Visualization:** Dominik Deffner.

**Writing – original draft:** Dominik Deffner.

**Writing – review & editing:** Richard McElreath.

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
