## [Decision Letter · Decision Letter 0]

15 Dec 2021

PONE-D-21-32591When does selection favor learning from the old? Social learning in age-structured populationsPLOS ONE

Dear author,

Thank you for submitting this serious and solid paper to PLOS ONE. After a delay due to difficulties in finding reviewers, I have now received two reviewer reports, both positive overall. The only substantial critique is raised by Reviewer 1's report. Please address it as fully as possible. If I am satisfied with the way this point is addressed I might accept the paper directly and not send it back. (Also, one small thing: please format the references section following standard guidelines and reread it carefully, right now the references are neither numbered nor ordered and there are typos, e.g. geocentracy for gerontocracy.)

Again, thank you for submitting your work to our journal. Sincerely, Olivier Morin

(The below are standard PLOS submission instructions)

Journal Requirements:

“This work has been funded by the Max Planck Society.”

Reviewers' comments:

Reviewer's Responses to Questions

**Comments to the Author**

1. Is the manuscript technically sound, and do the data support the conclusions?

Reviewer #1: Yes

Reviewer #2: Yes

2. Has the statistical analysis been performed appropriately and rigorously? 

Reviewer #1: N/A

Reviewer #2: Yes

3. Have the authors made all data underlying the findings in their manuscript fully available?

Reviewer #1: Yes

Reviewer #2: Yes

4. Is the manuscript presented in an intelligible fashion and written in standard English?

Reviewer #1: Yes

Reviewer #2: Yes

5. Review Comments to the Author

Reviewer #1: Dear Editor, dear authors,

This article entitled "When does selection favor learning from the old? Social learning in age-structured populations" models the evolution of learning strategies in an age-structured population and seeks to establish the conditions under which it is adaptive to learn preferentially from the old rather than the young.

The article is well written. The model is rigorous. The analyses are accurate. It is certainly an article that deserves to be published in PloS ONE.

However, I have one rather substantial criticism that I believe requires a significant correction to the article.

The most significant assumption of the model, which is very little discussed and yet has very important consequences, is the assumption that adults cannot learn.

As a result of this assumption, the only reason why old people may have better information than young people is because they have been more “tested” by mortality (demographic filtering).

However, this is only the case in this model, owing to the specific assumption that adults cannot learn.

In real life, demographic filtering is very unlikely to be the most general reason why old people may be better sources of information than young people. Old people are rather a better source of information because they have more experience. Having more experience has nothing to do with having survived longer. Having more experience is about having had more opportunities to learn, i.e. to have received more feedbacks from the environment. But this cannot be captured in a model where adults cannot learn. So the model fails to capture what is probably the most important and general reason why the old have better information than the young.

In itself, this is not a problem. It’s perfectly fine to build simplified models that capture only a part of reality.

However, the problem is that the authors do not acknowledge this limitation. In particular, they purport to relate their modeling results to empirical observations (lines 381-395), even though these observations are in fact very unlikely to be the consequence of demographic filtering, but much more likely to be consequence of differential experience. For example, if 15-month-olds are more likely to copy behaviors performed by an adult versus a two-year-old child (as cited by the authors, line 382), it is very likely NOT because adults have been tested by mortality, but more simply because they have had more opportunities to learn. And this is probably the same for all the empirical data cited by the author (lines 381-395).

So the model just superficially looks like it predicts empirical and real life observations while, in reality, it fails to capture the likely true cause of these observations (differential experience). The authors must make this point explicit. Otherwise they misled the readers into believing that they are reading a model that is much more interesting than it really is.

For example, the authors are extremely misleading in line 192 after their paragraph on empirical studies: "Taken together, these studies seem to imply that under most circumstances learners preferentially attend to older, more experienced, individuals. Our modeling results show that this can be a good strategy, but only when the environment is relatively stable, adaptive behavior is hard to acquire and confers large survival advantages." They present empirical observations that are caused by differential experience and claim that they are explained by their model that only captures demographic filtering!

In the article, the authors express their view on the right way to do modeling (line 334-337) a point on which I agree with them. Let me then add a short patronizing lesson. The aim of modeling is not to reproduce patterns that seem to resemble reality, without asking oneself whether the mechanism at the origin of these patterns is also in line with reality. Otherwise, modeling is nothing more than a magic trick. In a useful model, both the pattern produced AND the mechanism causing it must be in line with empirical data.

Small comments:

1) Regarding "age structure and Rogers' paradox". In the abstract, in the discussion, and also in the title of this section, the authors suggest that there is a real scientific question: "Does adding age structure resolve Rogers' paradox?".

This is rather artificial. There is nothing surprising about the fact that age structure does not resolve Rogers' paradox in the absence of a preferential transmission of accurate information. The authors should rather say something like: ”We replicate Rogers' paradox with age structure". That would be less misleading.

2) This sentence is not clear at all (line 359): “Even though social learning did not increase mean population fitness, in combination with demographic filtering, it enabled the population to reach much higher adaptation levels.”

Reviewer #2: First, I should say that I have no major critiques of the manuscript and I would be happy to see it published as-is. Both the mathematical and agent-based models are as simple as possible given the research questions, and the interpretation of their output appears to be sound. I only have a few minor recommendations:

- As far as I can tell "demographic filtering" is not defined in the paper, but appears throughout. It is worth defining this for readers (like myself) who are unfamiliar with the term.

- Part of the aims statement, on lines 56-58, is quite broad and hard to follow (for me, and possibly for other readers). It reads: "How do demographic and cultural forces interact in adaptation and population growth and what are the minimal conditions for such forces to make it worthwhile to copy older individuals?" I would consider making the "demographic and cultural forces interact in adaptation and population growth" part more specific, because right now I could think of many possible research questions that fit that description.

- It might be worth clarifying the description of Figure 1 on lines 160-162. I think the fact that the black shapes and colored lines represent two different sets of results is confusing on an intuitive level since they are in the same position (which I know is the point being made, but still might be worth clarifying).

- Have you considered including Figure S3 in the body text? I think it does an excellent job of communicating the influence of environmental stability, and would enhance readers' understanding of that result.

6. PLOS authors have the option to publish the peer review history of their article (what does this mean?). If published, this will include your full peer review and any attached files.

Reviewer #1: No

Reviewer #2: No

---

## [Author Response · Author response to Decision Letter 0]

24 Jan 2022

Responses to the reviewers are included as separate document

---

## [Decision Letter · Decision Letter 1]

5 Apr 2022

When does selection favor learning from the old? Social learning in age-structured populations

PONE-D-21-32591R1

Dear Dr. Deffner,

We’re pleased to inform you that your manuscript has been judged scientifically suitable for publication and will be formally accepted for publication once it meets all outstanding technical requirements.

Kind regards,

Olivier Morin

Academic Editor

PLOS ONE

Additional Editor Comments (optional):

Reviewers' comments:

Reviewer's Responses to Questions

**Comments to the Author**

1. If the authors have adequately addressed your comments raised in a previous round of review and you feel that this manuscript is now acceptable for publication, you may indicate that here to bypass the “Comments to the Author” section, enter your conflict of interest statement in the “Confidential to Editor” section, and submit your "Accept" recommendation.

Reviewer #1: All comments have been addressed

Reviewer #2: All comments have been addressed

2. Is the manuscript technically sound, and do the data support the conclusions?

Reviewer #1: Yes

Reviewer #2: Yes

3. Has the statistical analysis been performed appropriately and rigorously? 

Reviewer #1: N/A

Reviewer #2: Yes

4. Have the authors made all data underlying the findings in their manuscript fully available?

Reviewer #1: Yes

Reviewer #2: Yes

5. Is the manuscript presented in an intelligible fashion and written in standard English?

Reviewer #1: Yes

Reviewer #2: Yes

6. Review Comments to the Author

Reviewer #1: The authors have satisfyingly addressed my comments, especially in the discussion (lines 417-423), where they clearly state the fact that if people learn more from the old in real life it does not necessarily prove that demographic filtering is at work, as there are many other good reasons to prefer to learn from experienced people rather than from beginners.

I'm less convinced by the usefulness of their new model where adultes can learn (and beta varies). If I understand correctly they show that there can be old-bias even when adults are learning (which is obvious) whereas the reality (and what should have been tested) is that there can very likely be old-bias even without demographic filtering.

Reviewer #2: All of my minor recommendations have been adequately addressed, so I have no further comments for the authors.

7. PLOS authors have the option to publish the peer review history of their article (what does this mean?). If published, this will include your full peer review and any attached files.

Reviewer #1: No

Reviewer #2: No

---

## [Editor Report · Acceptance letter]

7 Apr 2022

PONE-D-21-32591R1 

When does selection favor learning from the old? Social Learning in age-structured populations 

Dear Dr. Deffner:

I'm pleased to inform you that your manuscript has been deemed suitable for publication in PLOS ONE. Congratulations! Your manuscript is now with our production department. 

Kind regards, 

on behalf of

Dr. Olivier Morin 

Academic Editor

PLOS ONE